



# Brief communication: CESM2 climate forcing (1950–2014) yields realistic Greenland ice sheet surface mass balance

**Brice Noël**[1], **Leonardus van Kampenhout**[1], **Willem Jan van de Berg**[1], **Jan T. M. Lenaerts**[2], **Bert Wouters**[1,3], **and Michiel R. van den Broeke**[1]

[1]Institute for Marine and Atmospheric research (IMAU), Utrecht University, Utrecht, the Netherlands
[2]Department of Atmospheric and Oceanic Sciences, University of Colorado Boulder, Boulder, CO, USA
[3]Department of Geoscience and Remote Sensing, Delft University of Technology, Delft, the Netherlands

**Correspondence:** Brice Noël (b.p.y.noel@uu.nl)

**Abstract.** We present a reconstruction of historical (1950–2014) surface mass balance (SMB) of the Greenland ice sheet (GrIS) using a high-resolution regional climate model (RACMO2; ~ 11 km) to dynamically downscale the climate of the Community Earth System Model version 2 (CESM2; ~ 111 km). After further statistical downscaling to 1 km spatial resolution, evaluation using in situ SMB measurements and remotely sensed GrIS mass change shows good agreement. Comparison with an ensemble of previously conducted RACMO2 simulations forced by climate reanalysis demonstrates that the current product realistically represents the long-term average and variability of individual SMB components and captures the recent increase in meltwater runoff that accelerated GrIS mass loss. This means that, for the first time, climate forcing from an Earth system model CE1 (CESM2), which assimilates no observations, can be used without additional corrections to reconstruct the historical GrIS SMB and its recent decline that initiated mass loss in the 1990s. This paves the way for attribution studies of future GrIS mass loss projections and contribution to sea level rise.

## 1 Introduction

A common approach to project the future surface mass balance (SMB) of the Greenland ice sheet (GrIS) is to force a regional climate model (RCM), typically running at 5 to 10 km horizontal resolution, at the lateral and top boundaries with the outputs of an Earth system model (ESM; ~ 100 km) (Van Angelen et al., 2013a; Fettweis et al., 2013; Mottram et al., 2017). However, ESMs from the fifth phase of the Climate Model Intercomparison Project (CMIP5) do not accurately represent the contemporary large-scale climate of the Greenland region (Rae et al., 2012; Fettweis et al., 2013). The reason is that ESMs neither assimilate nor prescribe climatic observations as global climate reanalyses (Uppala et al., 2005; Dee et al., 2011) and RCMs do (Fettweis et al., 2017; Mottram et al., 2017; Niwano et al., 2018; Noël et al., 2018). For instance, ESMs fail at capturing the recent summertime Arctic atmospheric circulation change (Hanna et al., 2018), making projections of GrIS mass loss and contribution to sea level rise highly uncertain (Delhasse et al., 2018). Consequently, climate forcing from CMIP5 ESMs still requires dedicated bias correction before being used to force RCMs over the GrIS (Rae et al., 2012; Fettweis et al., 2013; Van Angelen et al., 2013a). An alternative approach is to directly use outputs of ESMs to estimate GrIS SMB; however, most ESMs do not have (sophisticated) snow models that consider meltwater retention in firn, and their coarse spatial resolution does not accurately resolve the large SMB gradients at the GrIS margins (Lenaerts et al., 2019).

Here, we use the historical climate (1950–2014) of a CMIP6 model, the Community Earth System Model version 2.1 (CESM2; ~ 111 km), to force the lateral and top boundaries of the Regional Atmospheric Climate Model version 2.3p2 (RACMO2; ~ 11 km). The reason for selecting CESM2 as climate forcing for RACMO2 stems from the active involvement of the Institute for Marine and Atmospheric research Utrecht (IMAU) in the development and

improvement of the model for studies over both the Greenland and Antarctic ice sheets. To obtain a meaningful comparison with in situ observations, the resulting SMB field is then statistically downscaled to 1 km over the GrIS and peripheral glaciers and ice caps (Fig. 1a) (Noël et al., 2018). We show that, without additional corrections, CESM2 climate forcing yields a realistic reconstruction of historical GrIS SMB (1950–2014), including its recent decline in the 1990s. This is unexpected for an ESM which exclusively prescribes greenhouse gas ($CO_2$ and $CH_4$) and aerosol emissions and may herald more accurate projections of GrIS contribution to future sea level rise. Section 2 describes CESM2 and RACMO2, including model initialisation, forcing set-up, as well as observational and model data sets used for evaluation. Section 3 evaluates the CESM2-forced RACMO2 product using in situ and remotely sensed measurements. Model comparison to previous RACMO2 simulations is discussed in Sect. 4, as well as representation of recent trends in SMB components and mass loss. Conclusions are drawn in Sect. 5.

## 2 Methods

### 2.1 The Community Earth System Model: CESM2

CESM2.1, hereafter referred to as CESM2, is an ESM that simulates mutual interactions between atmosphere–ocean–land systems on the global scale. The model incorporates the Community Atmosphere Model version 6 (CAM6) (Gettelman et al., 2019), resolving global atmospheric dynamics and physics, the Parallel Ocean Program model version 2.1 (POP2.1) (Smith et al., 2010), and the Los Alamos National Laboratory Sea Ice Model version 5.1 (CICE5.1) (Bailey et al., 2018), modelling global oceanic circulation and sea-ice evolution. These are coupled with the Community Land Model version 5 (CLM5) (Lawrence et al., 2019) and the Community Ice Sheet Model version 2.1 (CISM2.1) (Lipscomb et al., 2019) simulating land–atmosphere interactions and ice dynamics. Here, we use a full atmosphere–ocean coupling in CESM2, i.e. including sea ice dynamics and sea surface temperature evolution while excluding land ice dynamics (e.g. calving). The model is run at 1° spatial resolution ($\sim 111$ km) and only prescribes atmospheric greenhouse gas ($CO_2$ and $CH_4$) and aerosol emissions as well as land cover use (Eyring et al., 2016). CESM2 has been extensively tested and adapted to realistically reproduce the contemporary climate and SMB of the GrIS (Van Kampenhout et al., 2017, 2019). Detailed model description, latest updates and evaluation are provided in Van Kampenhout et al. (2020).

### 2.2 Regional Atmospheric Climate Model: RACMO2

RACMO2 is an RCM that is specifically adapted to simulate the SMB of polar ice sheets (Noël et al., 2018; Van Wessem et al., 2018). The model incorporates the dynamical core of the High Resolution Limited Area Model (HIRLAM)

(Undèn et al., 2002) and the physics package cycle CY33r1 of the European Centre for Medium-Range Weather Forecasts Integrated Forecast System (ECMWF-IFS, 2008). It includes a multi-layer snow module that simulates melt, liquid water percolation and retention, refreezing and runoff (Ettema et al., 2010) and accounts for dry-snow densification following Ligtenberg et al. (2011). RACMO2 implements an albedo scheme that calculates snow albedo based on prognostic snow grain size, cloud optical thickness, solar zenith angle and impurity concentration in snow (Kuipers Munneke et al., 2011). In line with in situ measurements (Doherty et al., 2010), impurity concentration (soot) in RACMO2 is prescribed as a constant in time and space at 0.1 ppmv (Noël et al., 2018). The model simulates drifting snow erosion and sublimation following Lenaerts et al. (2012). The latest model version RACMO2.3p2 accurately simulates the contemporary climate and SMB of the GrIS when it is forced by ERA-40 (1958–1978) and ERA-Interim (1979–present) climate reanalyses (Uppala et al., 2005; Dee et al., 2011) and is statistically downscaled to 1 km spatial resolution (see Sect. 2.4). For detailed model description, latest updates and evaluation, we refer to Noël et al. (2018, 2019).

### 2.3 Model initialisation and set-up

Here, we conduct a CMIP6-style historical simulation (1950–2014) using RACMO2.3p2 at 11 km horizontal resolution (Noël et al., 2018) to dynamically downscale the outputs of CESM2 prescribed in a 24-grid-cell-wide relaxation zone at the model lateral boundaries. Forcing consists of atmospheric temperature, pressure, specific humidity, wind speed and direction being prescribed on a 6-hourly basis at the 40 model atmospheric levels. Upper atmosphere relaxation is implemented (Van de Berg and Medley, 2016). Sea surface temperature and sea ice extent/cover are prescribed from the CESM2 forcing every 6 h. RACMO2.3p2 has typically 40 to 60 active snow layers that are initialised in January 1950 using temperature and density profiles derived from the offline IMAU Firn Densification Model (IMAU-FDM) (Ligtenberg et al., 2018). Glacier outlines and surface topography are prescribed from a down-sampled version of the 90 m Greenland Ice Mapping Project (GIMP) digital elevation model (DEM) (Howat et al., 2014). Bare ice albedo is prescribed from the 500 m MODerate-resolution Imaging Spectroradiometer (MODIS) 16 d TS1 Albedo product (MCD43A3), as the 5 % lowest surface albedo records for the period 2000–2015, clipped between 0.30 for bare ice and 0.55 for bright ice covered by perennial firn in the accumulation zone. The current study uses the climate forcing of 1 out of the 12 members of the CESM2 historical ensemble. Forcing RACMO2 with other CESM2 members would have been ideal, but doing so in a transient fashion and at high spatial and temporal resolution is computationally prohibitive. Instead, we select one member that offers the 6-hourly climate

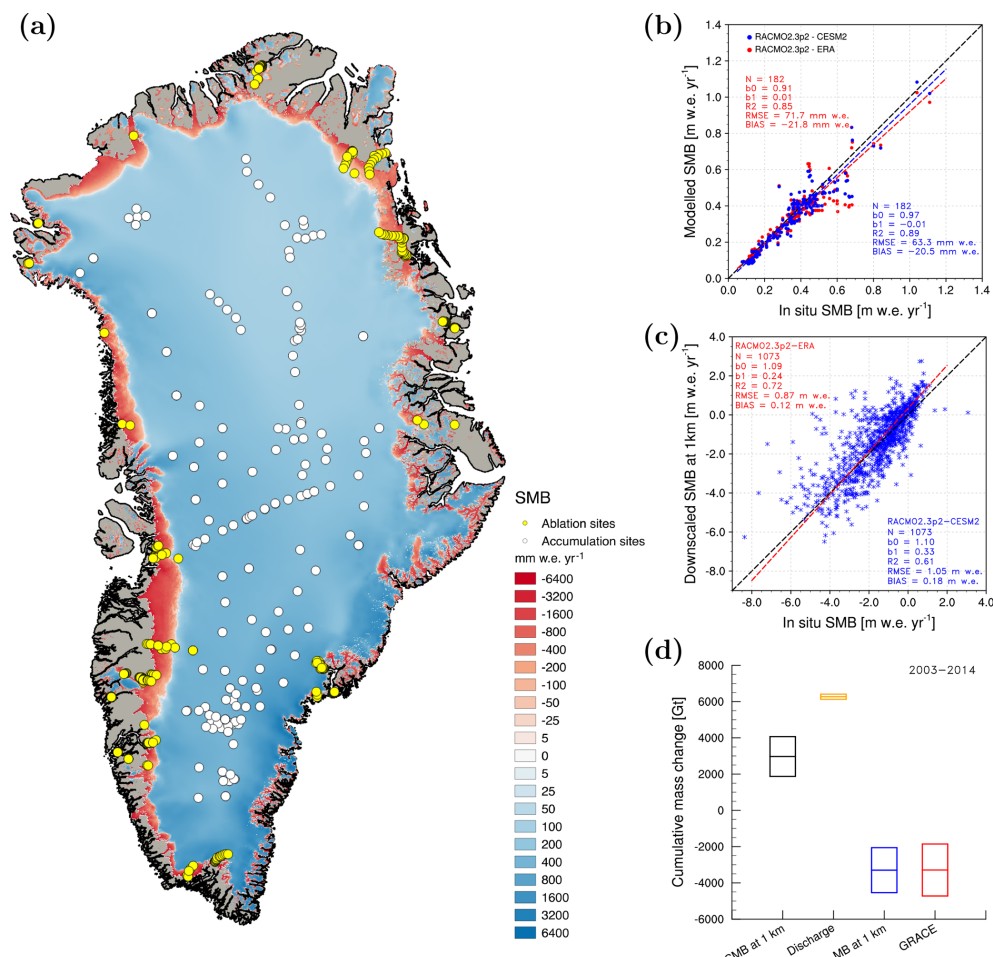

**Figure 1. (a)** Annual mean SMB (1950–2014) as modelled by RACMO2.3p2 forced by CESM2, statistically downscaled to 1 km resolution. **(b)** SMB evaluation in the accumulation zone (182 sites; white dots in **a**) and **(c)** in the ablation zone of the GrIS (213 sites; yellow dots in **a**). Statistics including the number of observations ($N$), slope ($b0$) and intercept ($b1$) of the regressions, determination coefficient ($R^2$), RMSE and bias are listed for the ERA (red) (Noël et al., 2018) and CESM2-forced RACMO2.3p2 simulation (blue). **(d)** Period (2003–2014) cumulative SMB (black), glacial discharge (orange) (Mouginot et al., 2019), mass balance (MB = SMB − discharge; blue) and mass loss derived from GRACE (red) (Wouters et al., 2013). To enable a direct comparison with GRACE in **(d)**, SMB is integrated over the GrIS, peripheral ice caps and tundra regions of Greenland. Boxes represent 1 standard deviation around the mean (horizontal bars).

forcing required to drive RACMO2 while being representative of other CESM2 members (see Sect. 4.3 and Fig. 4a).

## 2.4 Statistical downscaling

Following Noël et al. (2016), the historical simulation at 11 km, hereafter referred to as CESM2-forced RACMO2.3p2, is further statistically downscaled to a 1 km ice mask and topography derived from the 90 m GIMP DEM (Howat et al., 2014). In brief, the downscaling procedure corrects individual SMB components (except for precipitation), i.e. primarily meltwater runoff, for elevation and ice albedo biases on the relatively coarse model grid at 11 km resolution. These corrections reconstruct individual SMB components on the 1 km GrIS topography using daily-specific gradients estimated at 11 km and minimise the remaining runoff underestimation using a down-sampled 1 km MODIS 16 d ice albedo product averaged for 2000–2015. Precipitation, including snowfall and rainfall, is bilinearly interpolated from the 11 km onto the 1 km grid without additional corrections (Noël et al., 2018). Statistical downscaling proves essential to resolve narrow ablation zones, outlet glaciers and ice caps at the GrIS margins that significantly contribute to contemporary mass loss of Greenland land ice (Noël et al., 2017, 2019). For instance, applying statistical downscaling increases GrIS-wide runoff by 55 Gt yr$^{-1}$ (+23 %) on average for the period 1950–2014, resulting in a SMB decrease of 56 Gt yr$^{-1}$ (−13 %).

## 2.5 Evaluation data sets

For evaluation, we use a compilation of in situ SMB measurements derived from 182 stakes, snow pits (Bales et al., 2009) and airborne radar campaign (Overly et al., 2016) in the GrIS accumulation area (182 records; white dots in Fig. 1a) and collected at 213 sites in the ablation zone (1073 records; yellow dots in Fig. 1a) (Machguth et al., 2016). In addition, combined modelled SMB and glacial discharge estimates TS2 (Mouginot et al., 2019) are compared to mass changes from GRACE over the period 2003–2014 (Wouters et al., 2013). The CESM2-forced RACMO2.3p2 historical simulation is also compared to SMB and individual components from an ensemble of eight previous RACMO2 simulations (Van Angelen et al., 2013a, b; Noël et al., 2015, 2016, 2018, 2019), using different climate forcing (ERA reanalysis or the ESM HadGEM2) at various spatial resolutions (1, 5.5 and 11 km). These simulations are listed and further compared in Tables 1 and 2 for the overlapping model period 1960–2012. TS3

## 3 Surface mass balance evaluation and uncertainty

Figure 1a shows annual mean SMB from CESM2-forced RACMO2.3p2, statistically downscaled to 1 km. As is the case with state-of-the-art reanalysis-forced simulations (Mottram et al., 2017; Fettweis et al., 2017; Niwano et al., 2018; Noël et al., 2018, 2019), it accurately captures the extensive inland accumulation area as well as narrow ablation zones, outlet glaciers and ice caps fringing the GrIS margins (Fig. 1a). The model shows very good agreement with multi-year averaged SMB observations in the accumulation zone ($R^2 = 0.89$; Fig. 1b), with a small bias and RMSE of $-20.5$ and $63.3$ mm w.e. Interestingly, these statistics are on par with the recent RACMO2.3p2 run at 11 km forced by ERA reanalysis and statistically downscaled to 1 km (Noël et al., 2018), hereafter referred to as ERA-forced RACMO2.3p2. In the ablation zone, CESM2-forced RACMO2.3p2 agrees reasonably well with ablation measurements: $R^2 = 0.61$ vs. 0.72 (Noël et al., 2018) (Fig. 1c). The model shows larger bias and RMSE relative to ERA-forced RACMO2.3p2 ($+0.06$ and $+0.18$ m w.e.). As CESM2 neither assimilates nor prescribes climatic observations, a larger bias was expected. Good agreement with observations can be partly attributed to dynamical downscaling in RACMO2, which results in realistic SMB gradients if appropriate climate forcing is prescribed (Noël et al., 2018), and to statistical downscaling, as it minimises SMB bias by enhancing runoff in marginal ablation zones (Noël et al., 2016). On the regional scale, CESM2-forced and ERA-forced RACMO2.3p2 simulations show no significant difference in SMB and components for the period 1958–2014 (not shown), i.e. mean difference (CESM2-forced minus ERA-forced) lower than 1 standard deviation of the 1958–2014 period.

We follow Noël et al. (2017) to estimate the SMB uncertainty ($\sigma$). Mean accumulation (20.5 mm w.e.; Fig. 1b) and ablation biases (180.0 mm w.e.; Fig. 1c) are accumulated over the long-term (1958–2014) accumulation and ablation zone of the GrIS, with an area of $\sim 1\,521\,400$ and $\sim 179\,400$ km$^2$ respectively.

$$\sigma = \sqrt{(\text{bias}^{\text{abl.}} \times \text{area}^{\text{abl.}})^2 + (\text{bias}^{\text{acc.}} \times \text{area}^{\text{acc.}})^2} \quad (1)$$

CE2 This yields $\sigma = 48$ Gt yr$^{-1}$. A similar value (43 Gt yr$^{-1}$) is obtained for the downscaled ERA-forced RACMO2.3p2 product. Integrated over the ice sheet, CESM2-forced and ERA-forced simulations agree very well, with an average cumulative SMB of $365 \pm 48$ and $357 \pm 43$ Gt yr$^{-1}$ for the period 1958–2014. Figure 1d compares modelled and remotely sensed (GRACE) cumulative mass change for the period 2003–2014, respectively. Modelled mass change ($-3299 \pm 1240$ Gt; blue box), estimated as cumulative SMB over the GrIS, peripheral ice caps and tundra region ($2970 \pm 1097$ Gt; black box) minus glacial discharge ($6269 \pm 143$ Gt; orange box), shows excellent agreement with GRACE ($-3290 \pm 1434$ Gt; red box). This highlights the ability of CESM2-forced RACMO2.3p2 to also realistically capture the recent Greenland mass loss (2003–2014) (Bamber et al., 2018).

## 4 Surface mass balance variability

Figure 2a and c show annual mean SMB components simulated by CESM2-forced RACMO2.3p2 at 1 km (horizontal bars) for the periods 1960–1990 TS8 and 1991–2012. Likewise, Fig. 2b and d show annual mean mass balance (MB; blue), i.e. SMB (black) minus glacial discharge (orange), simulated by CESM2-forced RACMO2.3p2 on the original model grid (11 km) and statistically downscaled to 1 km for the two periods. Ice discharge and associated uncertainties (1972–2014) are from Mouginot et al. (2019). Prior to 1972, ice discharge and uncertainties are assumed constant at the 1972 value. Error bars represent the inter-annual variability in SMB components estimated as 1 standard deviation around the mean. Boxes show the range of modelled SMB components derived from an ensemble of seven RACMO2 simulations forced by ERA reanalysis. To highlight recent improvements in ESM climate forcing, we use the modelled SMB components from a previous RACMO2.1 simulation forced by HadGEM2 (dark green dots). HadGEM2 is a coupled atmosphere–ocean model developed by the Met Office Hadley Centre using a spatial resolution of $1.875° \times 1.25°$ (atmosphere) and $1° \times 1°$ (ocean). Similar to CESM2, the model prescribes land cover use, greenhouse gas and aerosol emissions and simulates the dynamic evolution of sea ice and sea surface temperature over the historical period (Jones et al., 2011). Compared to RACMO2.1, the latest RACMO2.3p2 version implements (1) significant changes in the cloud physics favouring more snowfall in the ice sheet

**Table 1.** Annual mean SMB and components integrated over the GrIS (Gt yr$^{-1}$) for the period 1960–1990 [TS4] from an ensemble of RACMO2 simulations using various spatial resolutions and lateral forcing. SMB components include total precipitation (PR), runoff (RU), melt (ME), refreezing (RF) and rainfall (RA). The uncertainty range corresponds to 1 standard deviation around the mean. Here mass balance of the GrIS (MB) is estimated as GrIS-integrated SMB minus glacial discharge for the period 1960–1990 [TS5] (458 Gt yr$^{-1}$) (Mouginot et al., 2019).

| 1960–1990 | Reference | Forcing | Grid | MB | SMB | PR | RU | ME | RF | RA |
|---|---|---|---|---|---|---|---|---|---|---|
| RACMO2.1 | Van Angelen et al. (2013a) | HadGEM2 | 11 km | $-189 \pm 97$ | $269 \pm 104$ | $685 \pm 82$ | $377 \pm 65$ | $604 \pm 116$ | $278 \pm 66$ | $51 \pm 15$ |
| RACMO2.1 | Van Angelen et al. (2013b) | ERA | 11 km | $-22 \pm 92$ | $437 \pm 99$ | $724 \pm 77$ | $247 \pm 47$ | $424 \pm 60$ | $220 \pm 27$ | $44 \pm 11$ |
| RACMO2.3p1 | Noël et al. (2015) | ERA | 11 km | $-45 \pm 91$ | $414 \pm 98$ | $693 \pm 72$ | $237 \pm 48$ | $404 \pm 62$ | $189 \pm 25$ | $22 \pm 6$ |
| RACMO2.3p2 | Noël et al. (2018) | ERA | 11 km | $28 \pm 93$ | $487 \pm 100$ | $715 \pm 73$ | $197 \pm 46$ | $345 \pm 58$ | $169 \pm 20$ | $23 \pm 6$ |
| RACMO2.3p2 | Noël et al. (2019) | ERA | 5.5 km | $41 \pm 96$ | $500 \pm 104$ | $731 \pm 75$ | $197 \pm 47$ | $341 \pm 59$ | $164 \pm 20$ | $23 \pm 6$ |
| RACMO2.3p2 | This study | CESM2 | 11 km | $26 \pm 99$ | $485 \pm 107$ | $702 \pm 98$ | $187 \pm 36$ | $332 \pm 66$ | $167 \pm 35$ | $23 \pm 7$ |
| RACMO2.3p1 | Noël et al. (2016) | ERA | 1 km | $-48 \pm 100$ | $411 \pm 108$ | $768 \pm 76$ | $313 \pm 55$ | $527 \pm 66$ | $240 \pm 28$ | $24 \pm 6$ |
| RACMO2.3p2 | Noël et al. (2018) | ERA | 1 km | $-36 \pm 98$ | $423 \pm 106$ | $712 \pm 73$ | $257 \pm 53$ | $457 \pm 56$ | $250 \pm 23$ | $35 \pm 6$ |
| RACMO2.3p2 | Noël et al. (2019) | ERA | 1 km | $-12 \pm 101$ | $447 \pm 109$ | $731 \pm 75$ | $250 \pm 53$ | $460 \pm 59$ | $253 \pm 21$ | $33 \pm 6$ |
| RACMO2.3p2 | This study | CESM2 | 1 km | $-31 \pm 100$ | $428 \pm 107$ | $701 \pm 98$ | $242 \pm 40$ | $457 \pm 63$ | $266 \pm 32$ | $37 \pm 8$ |

**Table 2.** Annual mean SMB and components integrated over the GrIS (Gt yr$^{-1}$) for the period 1991–2014 [TS6] from an ensemble of RACMO2 simulations using various spatial resolutions and lateral forcing. SMB components include total precipitation (PR), runoff (RU), melt (ME), refreezing (RF) and rainfall (RA). The uncertainty range corresponds to 1 standard deviation around the mean. Here mass balance of the GrIS (MB) is estimated as GrIS-integrated SMB minus glacial discharge for the period 1991–2012 (485 Gt yr$^{-1}$) (Mouginot et al., 2019).

| 1991–2012 | Reference | Forcing | Grid | MB | SMB | PR | RU | ME | RF | RA |
|---|---|---|---|---|---|---|---|---|---|---|
| RACMO2.1 | Van Angelen et al. (2013a) | HadGEM2 | 11 km | $-240 \pm 80$ | $245 \pm 141$ | $785 \pm 84$ | $496 \pm 114$ | $749 \pm 156$ | $318 \pm 54$ | $67 \pm 17$ |
| RACMO2.1 | Van Angelen et al. (2013b) | ERA | 11 km | $-155 \pm 80$ | $330 \pm 116$ | $731 \pm 64$ | $361 \pm 100$ | $588 \pm 146$ | $279 \pm 65$ | $52 \pm 14$ |
| RACMO2.3p1 | Noël et al. (2015) | ERA | 11 km | $-179 \pm 83$ | $306 \pm 119$ | $685 \pm 64$ | $336 \pm 99$ | $543 \pm 145$ | $233 \pm 60$ | $26 \pm 8$ |
| RACMO2.3p2 | Noël et al. (2018) | ERA | 11 km | $-95 \pm 78$ | $389 \pm 114$ | $709 \pm 61$ | $286 \pm 92$ | $470 \pm 133$ | $208 \pm 53$ | $26 \pm 8$ |
| RACMO2.3p2 | Noël et al. (2019) | ERA | 5.5 km | $-94 \pm 81$ | $391 \pm 117$ | $717 \pm 62$ | $291 \pm 91$ | $469 \pm 131$ | $203 \pm 53$ | $26 \pm 8$ |
| RACMO2.3p2 | This study | CESM2 | 11 km | $-127 \pm 47$ | $358 \pm 83$ | $704 \pm 70$ | $314 \pm 93$ | $495 \pm 131$ | $211 \pm 51$ | $32 \pm 14$ |
| RACMO2.3p1 | Noël et al. (2016) | ERA | 1 km | $-200 \pm 95$ | $285 \pm 131$ | $757 \pm 66$ | $428 \pm 109$ | $680 \pm 146$ | $283 \pm 53$ | $27 \pm 9$ |
| RACMO2.3p2 | Noël et al. (2018) | ERA | 1 km | $-170 \pm 85$ | $315 \pm 121$ | $705 \pm 61$ | $357 \pm 101$ | $590 \pm 136$ | $28 \pm 52$ [TS7] | $38 \pm 8$ |
| RACMO2.3p2 | Noël et al. (2019) | ERA | 1 km | $-157 \pm 85$ | $328 \pm 121$ | $717 \pm 63$ | $353 \pm 96$ | $594 \pm 131$ | $290 \pm 50$ | $36 \pm 8$ |
| RACMO2.3p2 | This study | CESM2 | 1 km | $-195 \pm 49$ | $290 \pm 85$ | $702 \pm 70$ | $380 \pm 101$ | $620 \pm 128$ | $302 \pm 47$ | $44 \pm 13$ |

interior; (2) lower impurity concentration in snow (soot) and smaller snow grain size, both reducing the previously underestimated snow albedo; (3) less-active snow drift erosion limiting the overestimated exposure of bare ice notably in the northeast of Greenland. For additional information about the HadGEM2-forced RACMO2.1 simulation and settings, we refer the reader to Van Angelen et al. (2013a); key differences between RACMO2.1, RACMO2.3p1 and p2 are discussed in Noël et al. (2015, 2018). Annual mean SMB components and corresponding inter-annual variability for the ensemble RACMO2 simulations are listed in Tables 1 (1960–1990) [TS9] and 2 (1991–2012).

## 4.1 Approximate mass balance: 1960–1990 [TS13]

In the period 1960–1990 [TS14], the mass balance of the GrIS was close to zero (Van den Broeke et al., 2016) or slightly negative (Mouginot et al., 2019). Figure 2a, b and Table 1 show that downscaled CESM2-forced RACMO2.3p2 reproduces, within 1 standard deviation, SMB and components obtained from seven previous reanalysis-forced RACMO2 simulations at various spatial resolutions. For instance, precipita-

tion ($701 \pm 98$ Gt yr$^{-1}$) and runoff ($242 \pm 40$ Gt yr$^{-1}$) compare well with ERA-forced RACMO2.3p2 (Noël et al., 2018), i.e. $712 \pm 73$ and $257 \pm 53$ Gt yr$^{-1}$, resulting in similar SMB of 428 and 423 Gt yr$^{-1}$ ($-1$ %) (Fig. 2a and Table 1). This highlights the ability of the CESM2 forcing to capture realistic Greenland SMB before mass loss started in the 1990s.

Figure 2b shows that SMB on the 11 km grid falls well within ERA-forced simulations at similar resolution (black box). Through statistical downscaling, SMB at 1 km decreases by 13 % from 485 to 428 Gt yr$^{-1}$, in line with other simulations (Fig. 2b and Table 1). Combining average GrIS-integrated SMB with glacial discharge (458 Gt yr$^{-1}$; 1960–1990 [TS15]), CESM2-forced RACMO2.3p2 results in slightly negative mass balance ($-31$ Gt yr$^{-1}$; Table 1). A previous attempt using RACMO2.1 forced by the climate of HadGEM2 (Van Angelen et al., 2013a) did not accurately represent GrIS-integrated SMB components (dark green dots in Fig. 2a, b). While precipitation was generally well represented ($685 \pm 82$ Gt yr$^{-1}$), runoff was overestimated by $\sim$ 50 % compared to ERA-forced RACMO2.3p2 (Table 1). As a result, SMB was underestimated by $\sim$ 40 %, driving an

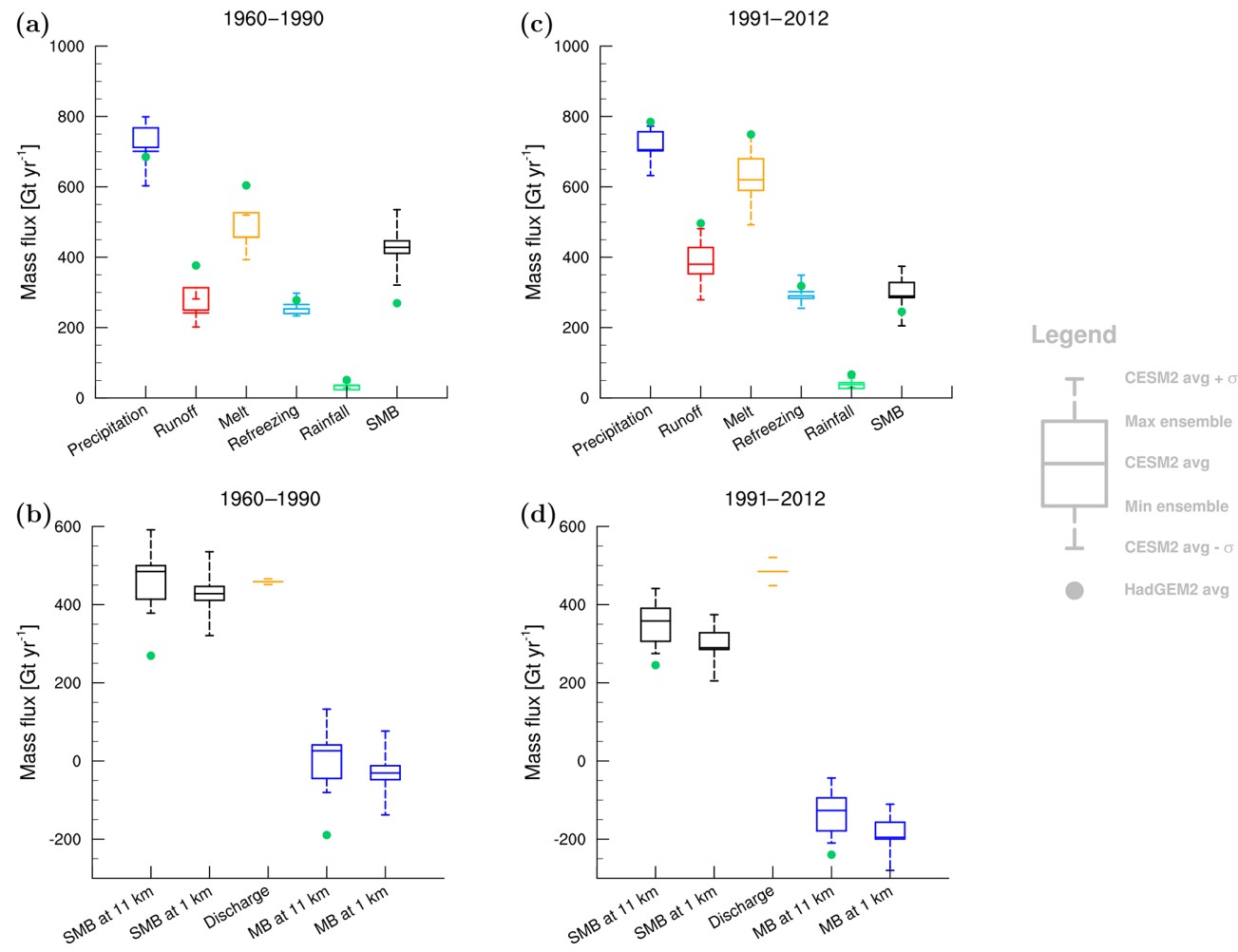

**Figure 2.** TS10 **(a)** Average GrIS SMB (black) and components at 1 km, i.e. total precipitation (blue), runoff (red), melt (orange), refreezing (cyan) and rainfall (green), for the period 1960–1990 TS11. **(b)** Annual mean SMB (black), glacial discharge (orange) (Mouginot et al., 2019) and mass balance (MB = SMB − discharge; blue) at 11 and 1 km resolution for the period 1960–1990 TS12. **(c** and **d)** same as (**a** and **b)** for the period 1991–2012. The green dots represent values from a previous HadGEM2-forced RACMO2.1 simulation. Boxes around the CESM2-forced RACMO2.3p2 mean (horizontal bars) represent the range of modelled estimates from an ensemble of RACMO2 simulations (five at 11 km, one at 5.5 km, and four at 1 km; Tables 1 and 2). The error bars represent 1 standard deviation ($\sigma$) around the CESM2-forced RACMO2.3p2 mean.

unrealistic mass loss of 189 Gt yr$^{-1}$ over the period 1960–1990 TS16.

### 4.2 Mass loss: 1991–2012

In the two decades following 1990 (1991–2012), the GrIS experienced accelerated mass loss (Bamber et al., 2018; The IMBIE team, 2019), primarily driven by a decrease in SMB (−138 Gt yr$^{-1}$ with respect to 1960–1990 TS17; Fig. 2c and Table 2) combined with an increase in glacial discharge (+26 Gt yr$^{-1}$). Figure 2c shows that CESM2-forced RACMO2.3p2 similarly reproduces the recent SMB decrease resulting from enhanced surface runoff (+138 Gt yr$^{-1}$ or +57 %) compared to the ERA-forced RACMO2.3p2 simulation (+100 Gt yr$^{-1}$ or +38 %). This pronounced runoff in-

crease stems from enhanced surface melt (+163 Gt yr$^{-1}$ or +36 %) exceeding the increase in meltwater retention and refreezing in the firn (+36 Gt yr$^{-1}$ or +14 %). Precipitation does not substantially change after 1991, in line with the ensemble ERA-forced RACMO2 simulations (Table 2). We conclude that CESM2-forced RACMO2.3p2 captures the post-1990 SMB decrease that tipped the GrIS into a state of sustained mass loss (195 and 170 Gt yr$^{-1}$ for downscaled CESM2-forced and ERA-forced RACMO2.3p2 for 1991–2012; Fig. 2d). In contrast, SMB components in HadGEM2-forced RACMO2.1 remain largely overestimated compared to other simulations (Table 2), particularly runoff and melt (Fig. 2c), resulting in overestimated mass loss for 1991–2012 (240±80 Gt yr$^{-1}$; Fig. 2d). The reason is that, unlike CESM2

(Van Kampenhout et al., 2020), the HadGEM2 forcing had a strong, systematic warm bias of $\sim 1\,°C$ (Van Angelen et al., 2013a), resulting in overestimated meltwater runoff and thus underestimated SMB (Fig. 2d).

## 4.3   Time series and trends

Figure 3a and b show time series of individual GrIS-integrated SMB components for the period 1950–2014 as modelled by the latest, state-of-the-art ERA-forced RACMO2.3p2 run at 5.5 km horizontal resolution (Noël et al., 2019) and the current CESM2-forced RACMO2.3p2 simulation, both statistically downscaled to 1 km. It is important to note that, compared to forcing by reanalyses that assimilate observations, the CESM2-forced simulation produces extreme melt years (e.g. 2005 and 2011; Fig. 3b) that are realistic in magnitude but not necessarily in timing (e.g. the observed 2012 melt peak; Fig. 3a). For 1960–1990, the two products show similar and insignificant trends in total precipitation ($2.4 \pm 1.6\,Gt\,yr^{-2}$; $p$ value $= 0.14$) and runoff ($1.1 \pm 1.0\,Gt\,yr^{-2}$; $p$ value $= 0.27$) (Fig. 3b). After 1991, CESM2-forced RACMO2.3p2 reproduces the significant ($p$ value $= 0.0001$) positive runoff trend ($10.4 \pm 2.2\,Gt\,yr^{-2}$; Fig. 3b) similar to the ERA-forced simulation ($8.8 \pm 2.1\,Gt\,yr^{-2}$ TS18; Fig. 3a). The runoff trend in CESM2-forced RACMO2.3p2 is no coincidence. Figure 4a shows the atmospheric temperature at 700 hPa ($T_{700}$) from the current CESM2 simulation (red) and from 11 additional ensemble members (grey). Compared to $T_{700}$ derived from ERA-40 (1958–1978) and ERA-Interim (1979–2014; black line in Fig. 4a), the current CESM2 simulation shows a cold bias of $0.6\,°C$ over 1958–2014. For the ERA-Interim period, the bias decreases to $0.4\,°C$. All CESM2 members show a similar warming trend after 1991, in line with the reanalysis data (dashed black line), highlighting the ability of CESM2 to represent the recent climate of Greenland. As in Fettweis et al. (2013), we find a clear correlation ($r = 0.67$; Fig. 4b) between CESM2-forced RACMO2.3p2 runoff at 1 km and $T_{700}$ from the CESM2 simulation (red). This means that the post-1990 runoff increase would have been obtained irrespective of the selected CESM2 member. Physical drivers of the warming trend in the CESM2 forcing are currently being investigated and will be discussed in a forthcoming publication. Compared to the ERA-forced run, the more pronounced runoff trend in CESM2-forced RACMO2.3p2 results from a significant ($p$ value $= 0.016$) positive trend in rainfall ($1.3 \pm 0.3\,Gt\,yr^{-2}$ vs. $0.3 \pm 0.2\,Gt\,yr^{-2}$) for a similar melt acceleration ($11.9 \pm 3.0\,Gt\,yr^{-2}$ vs. $10.9 \pm 3.0\,Gt\,yr^{-2}$). Total precipitation in the CESM2-forced RACMO2.3p2 simulation shows a significant ($p$ value $= 0.002$) positive trend ($5.8 \pm 1.7\,Gt\,yr^{-2}$; Fig. 3b) in contrast to a negative trend in the ERA-forced run ($-1.9 \pm 1.8\,Gt\,yr^{-2}$; Fig. 3a). However, the latter trend stems from decadal variability as it becomes insignificant for the period 1950–2014: $0.9 \pm 0.5\,Gt\,yr^{-2}$ ($p$ value $= 0.090$). In addition, the positive precipitation trend

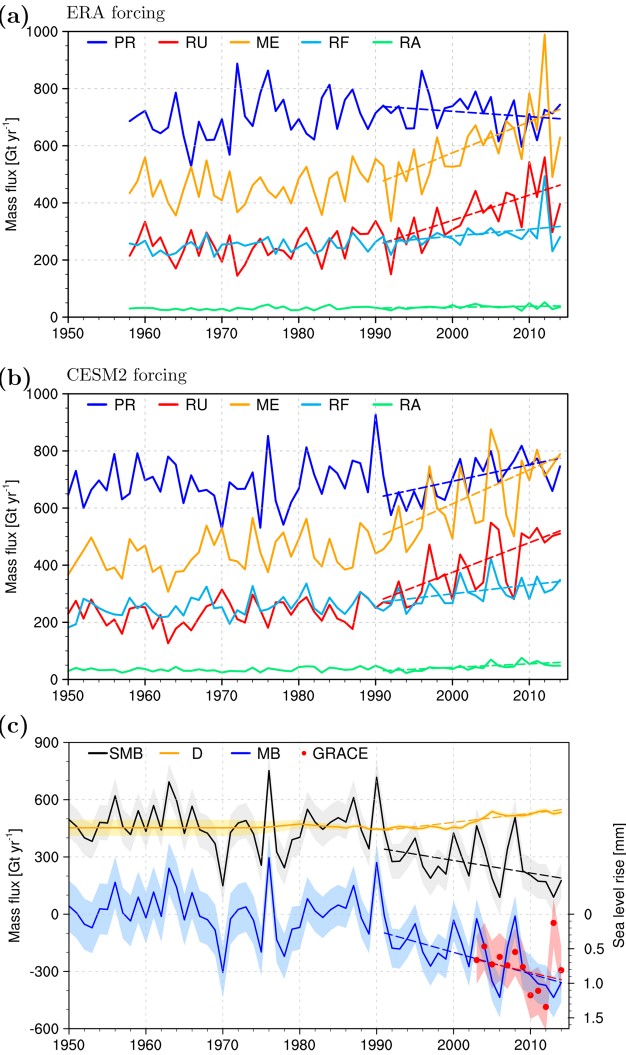

**Figure 3.** Time series of downscaled (1 km) GrIS-integrated annual SMB components, namely total precipitation (PR; blue), runoff (RU; red), melt (ME; orange), refreezing (RF; cyan) and rainfall (RA; green), as modelled by **(a)** ERA-forced RACMO2.3p2 (1958–2014) (Noël et al., 2019) and **(b)** CESM2-forced RACMO2.3p2 (1950–2014). **(c)** Time series of annual SMB (CESM2-forced RACMO2.3p2 at 1 km), glacial discharge ($D$) (Mouginot et al., 2019) and mass balance (MB $=$ SMB $- D$). Mass loss from GRACE (2003–2014) is represented by red dots (Wouters et al., 2013). Dashed lines show the 1991–2014 trends. To enable a direct comparison with GRACE in **(c)**, SMB is integrated over the GrIS, peripheral ice caps and surrounding tundra regions of Greenland.

disappears when extending time series using a CESM2-based SSP5-8.5 scenario (not shown), demonstrating that the latter trend originates from internal decadal variability.

In line with Van den Broeke et al. (2016), Fig. 3c shows that $\sim 60\%$ of the recent mass loss acceleration in CESM2-forced RACMO2.3p2 is caused by decreased SMB ($6.6 \pm 3.3\,Gt\,yr^{-2}$) resulting from enhanced meltwater runoff; the

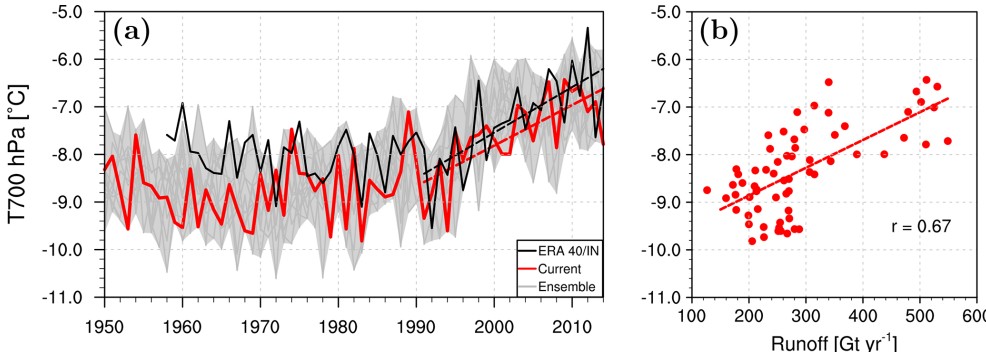

**Figure 4. (a)** Time series of the annual June–July–August (JJA) atmospheric temperature at 700 hPa ($T_{700}$) averaged over 60–80° N and 20–80° W for the ERA-40 (1958–1978) and ERA-Interim (1979–2014) reanalyses (black), the current CESM2 simulation (red) and 11 additional reference ensemble members (dark grey). The grey belt encompasses annual minimum and maximum values of the whole ensemble. **(b)** Annual GrIS-integrated runoff derived from CESM2-forced RACMO2.3p2 at 1 km resolution as a function of JJA atmospheric temperature at 700 hPa from the current CESM2 simulation (red).

remaining $\sim 40\%$ is ascribed to increased glacial discharge ($4.7 \pm 0.5\,\mathrm{Gt\,yr^{-2}}$). As a result, Greenland mass balance decreased by an estimated rate of $11.3 \pm 3.2\,\mathrm{Gt\,yr^{-2}}$ (or $9.4 \pm 1.6\,\mathrm{Gt\,yr^{-2}}$ for the GrIS only) in good agreement with GRACE ($9.4 \pm 1.2\,\mathrm{Gt\,yr^{-2}}$ for 2003–2014; Fig. 3c). It is important to note that the main drivers of the post-1990 (surface) mass loss in CESM2-forced RACMO2.3p2 may differ from that of the reanalysis-based products since ESMs from CMIP5 (and likely CMIP6) do not accurately reproduce the change in summertime Arctic atmospheric circulation that is often associated with the recent mass loss acceleration (Hanna et al., 2018). In brief, the current study is meaningful for two reasons: for the first time, an ESM, assimilating no observational climatic data except for atmospheric greenhouse gas and aerosol emissions, can (1) reliably reproduce the historical average and variability of SMB and its individual components and (2) realistically represent the recent Greenland mass loss acceleration in line with remote sensing. These results are essential for forthcoming attribution studies investigating post-1990 GrIS mass loss.

## 5   Conclusions

Historical outputs (1950–2014) of the Earth system model CESM2 ($\sim 111$ km) are dynamically downscaled using the regional climate model RACMO2.3p2 ($\sim 11$ km) over the GrIS. The resulting SMB components are further statistically downscaled to 1 km spatial resolution to resolve the narrow ablation zones and marginal outlet glaciers of Greenland. Model evaluation using in situ and remotely sensed measurements demonstrates the ability of CESM2-forced RACMO2.3p2 to realistically represent SMB as well as the rapid post-1990 melt and runoff increase. Combining modelled SMB with observed glacial discharge, our new ESM-based SMB product reflects an ice sheet in approximate mass balance before 1991, followed by a rapid mass loss accel-

eration resulting from enhanced meltwater runoff: two key features that, until now, exclusively showed up in reanalysis-based estimates. It is important to note that the main drivers of runoff increase in CESM2-forced RACMO2.3p2 may differ from that of reanalysis-based products since ESMs from CMIP5 (and likely CMIP6) do not accurately reproduce the recent change in summertime Arctic atmospheric circulation that is often associated with mass loss acceleration. For the first time, an Earth system model (CESM2), which does not assimilate climatic observations, can be used to force a regional climate model (RACMO2) to yield realistic historical GrIS SMB average and variability. Furthermore, our results suggest that CESM2 climate forcing can be used without bias corrections to project the SMB of the GrIS under different warming scenarios and quantify Greenland's contribution to future eustatic sea level rise.

*Data availability.* Data sets presented in this study are available from the authors upon request and without conditions.

*Author contributions.* BN prepared the manuscript, conducted the CESM2-forced RACMO2.3p2 simulation and analysed the data. LvK and JTML provided the historical CESM2 forcing. BW processed the GRACE mass anomalies time series. MRvdB and WJvdB helped interpreting the results. All authors commented the manuscript.

*Competing interests.* The authors declare that they have no conflict of interest.

*Acknowledgements.* Brice Noël was funded by NWO VENI grant VI.Veni.192.019. Leonardus van Kampenhout, Willem Jan van de Berg and Michiel van den Broeke acknowledge funding from

NWO and NESSC. Bert Wouters was funded by NWO VIDI grant 016.Vidi.171.063. The Community Earth System Model (CESM) project is supported primarily by the National Science Foundation (NSF). This material is based upon work supported by the National Center for Atmospheric Research (NCAR), which is a major facility sponsored by the NSF under cooperative agreement no. 1852977.

*Financial support.* This research has been supported by the Polar Programme of the Netherlands Organisation for Scientific Research (NWO) ((NWO) VENI (grant no. VI.Veni.192.019)).

*Review statement.* This paper was edited by Xavier Fettweis and reviewed by three anonymous referees.

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

TS18    Please explain this requested change in detail.