# Peer review of "Brief communication: CESM2 climate forcing (1950-2014) yields realistic Greenland ice sheet surface mass balance"

_The Cryosphere, 2019_

## Referee Comment (RC1) · Anonymous Referee #1 · 18 Oct 2019

Remarks to the Authors

Review of "Brief communication: CESM2 climate forcing (1950-2014) yields realistic Greenland ice sheet surface mass balance" by Brice Noël et al.

The Cryosphere Discuss. Manuscript Number: tc-2019-209
* * *
General comments:

In this paper, the authors introduce present-day historical (1950-2014) global model simulation data generated by the Community Earth System Model version 2 (CESM2),

which can be utilized to force polar regional climate models (RCMs) like RACMO2 used in this study. If "stand-alone" CESM2 can provide realistic climate forcing data for polar RCMs, it means that such RCMs are allowed to conduct a seamless model calculation from past to the present and future without any bias corrections. This kind of seamless simulation by a polar RCM is a state-of-the-art challenge, so that it can provide more realistic information related to possible future changes in the physical conditions of polar ice sheets as well as terrestrial climate system (e.g., sea level rise). Here, the authors perform dynamical down-scaling of the CESM2 data using RACMO2 in the Greenland ice sheet (GrIS) and try to prove the effectiveness of CESM2 through validating GrIS SMB simulated by RACMO2 (equipped with the statistical down-scaling postprocessing). This reviewer thinks that this considerable challenge is deserved to be published in the journal The Cryosphere as a brief communication if it is addressed in an appropriate manner. Overall, this paper is well written and structured; however, this reviewer suggests the following points to be considered before the publication.

Please note that page and line numbers are denoted by "P" and "L", respectively.

––––––––––––––––––––––––––––––––––––––––––––––––––––––––––––––––––––––

Specific comments (major)

P. 1, L. 9 $\sim$ 10 (and Sect. 2.1): According to the paper by Van Kampenhout et al. (2019b), which I read before reviewing this manuscript, the CESM2 simulation by Van Kampenhout et al. (2019b) was conducted following the so-called AMIP (Atmospheric Model Intercomparison Project)-run procedure. Did the authors use the same procedure/data as those presented by Van Kampenhout et al. (2019b)? If YES, the authors cannot argue "This means that, for the first time, an Earth System Model (CESM2), without assimilating observations, can be used to reconstruct historical GrIS SMB and the mass loss acceleration that started in the 1990s." in my opinion. It is because Van Kampenhout et al. (2019b) prescribe ocean and sea ice data at monthly intervals in their CESM2 simulation following the AMIP protocol, which is a kind of observation

data assimilation (I mean observed ocean physical conditions can drive changes in atmospheric conditions in the model, although the atmosphere-ocean interaction would not be so strong in the model). If NO, there is no doubt that this study is amazing, and I would like to congratulate for the achievement. Anyway, please clarify this point in Sect. 2.1.

P. 6, L. 32 ∼ P. 7, L. 2: Why does CESM2-forced RACMO2.3p2 show the significant positive trend of total precipitation since 1990, which is not shown in the ERA-forced run? Please discuss.

——————————————————————————————————————————

Specific comments (minor)

P. 2, L. 24: If possible, please indicate/mention the GrIS ice discharge simulated by the ice sheet model CISM2.1 incorporated in CESM2, which might be of interest to readers of The Cryosphere.

P. 2, L. 25 ∼ 26: Please indicate data sources for "atmospheric greenhouse gas emissions (CO2 and CH4), aerosol concentrations, and land cover use".

P. 3, L. 4: What kind of impurities do the authors consider in RACMO2 applied in the GrIS? And, how do the authors give concentrations of the impurities in the model?

P. 3, L. 15: Please indicate data source of sea surface temperature and sea ice extent used here (maybe from the parent CESM2 simulation results?).

P. 3, L. 19 ∼ 21: This sentence is a bit difficult to understand to me. Does it mean the 5 % lowest bare ice albedo from MCD43A3 is 0.30?

P. 5, L. 30 ∼ P. 6, L. 1: I think the results from RACMO2.1 forced by HadGEM2 is not necessary in this paper, because it can confuse readers who do not know much about RACMO2. If the authors think this part is really important for this paper, they should at least indicate key differences between RACMO2.1 and RACMO2.3p2 briefly in Sect.

2.2. Also, brief introduction of HadGEM2 would be needed as well.

P. 6, L. 15 ∼ 16: Same as the above comment.

P. 6, L. 24: Please indicate quantitatively how realistic the simulated T700 is.

P. 6, L. 28 ∼ 29: This reviewer agrees with the authors' point that the attempt mentioned here is very interesting as long as the CESM2 data used in this study is not from the AMIP-type simulation. Please also see my first major comment.

Figure 3: Can the authors briefly comment on why CESM2-forced RACMO2.3p2 could not simulate the 2012 extreme melt, which is simulated successfully by the ERA-forced run? I think this point is related to "physical drivers of the warming" (P. 6, L. 28), and any comments/suggestions by the authors will be informative for readers.

---

## Referee Comment (RC2) · Anonymous Referee #2 · 11 Nov 2019

**Review of Noel et al**

The authors present the results of one dynamically downscaled Earth System Model (ESM) simulation over the Greenland Ice Sheet (GrIS) and present the resulting historical surface mass balance (SMB) output from their regional climate model RACMO. After dynamical downscaling of the ESM input, the SMB is furthermore statistically downscaled to a nominal horizontal resolution of 1km.

**Overall Impression (Text and Figures):**
In general, the authors are doing a very good job in keeping their sentence and paragraph structure easy to follow and all their figures are well presented. Therefore, the manuscript is good to read.

**Scientific assessment:**
Overall, it's hard to make a case for how the study in its present form will benefit the wider cryospheric and climate community. The point of the authors here is to create a scientific foundation for additional papers that they want to write on the future contribution of the GrIS to sea level rise via (surface) mass loss. Overall, 21$^{st}$ century simulations of the GrIS climate and SMB would be very beneficial for the community, however, the presented analysis currently lacks the needed depth to be considered a valuable contribution to the field. Therefore, I would encourage the authors to consider the following points.

(1) The authors present only one RCM simulation forced with one GCM/ESM run to create a foundation for a future paper on 21$^{st}$ century GrIS climate projections. However, in its current form, the paper lacks (a) a consideration of the inter-model spread between all of the different GCMs in the CMIP5/6 model domain (b) a consideration of how the authors made their specific selection for the one run they choose out of their CESM2 ensemble. Fettweis et al (2013) for example analyse all the CMIP5 models over the current climate, selectively find the most suitable boundary forcings and create a downscaled RCM ensemble for multiple emission scenarios and models. This point is unfortunately omitted in this study.

(2) The authors focus their analysis only on the GrIS surface mass balance. If this study should become a standalone piece of work without the promised future projections, then the authors should be highly encouraged to consider at least a subset of other parameters to validate their single-simulation analysis to exclude the likelihood of compensating biases leading to a "correct" SMB due to "false" physical reasons - (a) Surface energy budget vs. observations (b) Albedo vs. observations (c) Temperature and/or cloud properties vs. observations.

(3) If the reader assesses the novelty based on what the authors highlight "…for the first time an ESM (CESM2) can be used to reconstruct historical SMB…" then the science of the paper would need to be judged either on the on the claim that is "the first time" or that the "historical SMB" is more accurate than from other model setups.

However, (a) e.g. Fettweis et al. (2013) as a benchmark already show that GCMs/ESMs can be used to force RCMs over the historical period and roughly get the magnitude of the SMB components right.

(b) The most accurate "historical SMB" does not come from this model setup, but rather from regional climate models that downscale observation-based reanalysis data (e.g.RCM with ERA-I or ERA-5). The presented results (Figure 3) unsurprisingly show that CESM2-RACMO does not capture the interannual SMB variability and extremes (e.g. melt in 2012) which is expected with GCM boundary forcings. However, it means that the accuracy of historical SMB representation is also not an advancement of the scientific knowledge.

**Recommendations:**

The reviewer would like to encourage the authors to either add significant extra analysis to their current model and study setup to create a solid foundation for their promised future attribution studies, or potentially add the presented analysis to their upcoming future projections altogether.

The authors could potentially consider some of the following points/questions when considering the next steps for their analysis post-review.

(a) Given the limited amount of future GrIS mass loss studies with RCMs and GCM forcing, the scientific interest of the presented approach lies in the actual future projections, not necessarily on the historical SMB reconstructions due to obvious limitations when using GCM/ESM boundary conditions.

(b) How representative is this one CESM2 run compared to the spread in CMIP5/6 simulations? Other recent studies have found great uncertainties in future GrIS projections using RCMs to downscale GCMs/ESMs which is/are not really discussed yet in the manuscript. What if the authors would force RACMO with other GCMs? How well does the current setup represent the surface energy budget, temperature, albedo, cloud properties?

(c) If forcing RACMO with other GCMs is technically not feasible, then one approach would be to force RACMO with additional ensemble members presented in Figure 4. The robustness of the SMB and potential underlying compensating errors can hardly be assessed by only one simulation.

**Minor comments:**

**P1.L9:** "without assimilating observations" is this correct? The methods of the paper claim that RACMO uses satellite albedo to constrain the surface albedo. Please clarify.

**P3.L19**: "bare ice albedo is prescribed from … MODIS.." – please see first minor comment and clarify.

**P3.L28** Also in the statistical downscaling technique the authors use observed MODIS albedo. Please see the first comment on how this fits with the claim that this study doesn't use assimilated observations.

**P3.L32-33**: Does it only *change* the runoff and SMB or also *improve* the statistical comparison?

**P4.L24**: "due to the high quality of the CESM2 climate" but also e.g. **P1.L5** "good comparison" and **P5.L6** "shows excellent agreement" and at other points in the manuscript - these are quite colloquial expressions with little scientific meaning. What does a "high quality" climate in a GCM mean? The manuscript doesn't even currently evaluate the CESM2 climate for example.

**P4.L25ff**: But what about other parameters such as the surface energy budget, temperature and clouds? How does it compare to recent circulation and cloud anomalies over Greenland which have been shown to be important for future projections?

**P5.L6-8:** The acceleration (i.e. dSMB/dt) is likely not discussed here but rather a "total mass loss".

**P5.L30-32ff:** ad HadGEM; "did not accurately reproduce SMB". Throughout this study the reader is often left in the dark as to "Why?" certain numbers or results are mentioned, and why certain processes behave the way they do. At the moment, the paper is an ensemble of nice figures and easy-to-follow text, but the study and the reader would highly benefit if the authors would more often dig into the question of "Why?" some processes and numbers are reported here and apparently deemed important for the reader. This would also be a good point to address the matter why HadGEM and CESM2 produce such different SMB/ME/RU results (+-50%)? Is it due to differences in the lateral forcings/ the internal RACMO physics/ circulation / cloud physics? Hofer et al. (2019) for example show the large spread in GrIS SMB that can result from different GCM forcing.

**P7.L8-9** "can reliably reproduce … variability of historical SMB" – When looking at Figure 3 the GCM-forced SMB reconstruction clearly lacks the ability to reproduce the interannual SMB variability and extremes shown in Figure 3A when RACMO is forced by reanalysis. Just as an example, the extreme melt summer of 2012 accurately captured in Figure 3A is not present in Figure 3B, therefore the the reader considers this to be a doubtful assumption.

P7.L3-4: unclear phrasing "is for 60%"

**P7.L7-10**: What are the uncertainties coming from the lack of a multi model forcing (e.g. Fettweis et al. (2013).

**Figure 1**: How does it compare during melt season? How does the SEB compare to the observational networks of PROMICE, DMI and/or GCNET?

**Figure 2**: Please clarify the choice of HadGEM and not other GCMs? If it is feasible to force RACMO with other GCMs then please consider analysing the intermodel spread of the GrIS climate when RACMO is forced by other GCMs

References

Fettweis, Xavier, et al. "Estimating Greenland ice sheet surface mass balance contribution to future sea level rise using the regional atmospheric climate model MAR." Cryosphere discussions 6 (2012): 3101-3147.

Hofer, Stefan, et al. "Cloud microphysics and circulation anomalies control differences in future Greenland melt." Nature Climate Change 9.7 (2019): 523.

---

## Author Comment (AC1) · 15 Jan 2020

Please, find attached our response to the reviewers and revised manuscript.

Please also note the supplement to this comment:
https://www.the-cryosphere-discuss.net/tc-2019-209/tc-2019-209-AC1-supplement.zip

---

## Author Response (AR1)

**Response to editor and reviewers:**

Dear editor and reviewers, we would like to thank you for your comments on our manuscript. To facilitate readability, our responses to reviewers are displayed in blue and modifications in the manuscript are highlighted in red. These suggested changes, together with additional minor corrections, are also displayed in red in the attached revised manuscript.

**Reviewer #1**

In this paper, the authors introduce present-day historical (1950-2014) global model simulation data generated by the Community Earth System Model version 2 (CESM2), which can be utilized to force polar regional climate models (RCMs) like RACMO2 used in this study. If "stand-alone" CESM2 can provide realistic climate forcing data for polar RCMs, it means that such RCMs are allowed to conduct a seamless model calculation from past to the present and future without any bias corrections. This kind of seamless simulation by a polar RCM is a state-of-the-art challenge, so that it can provide more realistic information related to possible future changes in the physical conditions of polar ice sheets as well as terrestrial climate system (e.g., sea level rise). Here, the authors perform dynamical downscaling of the CESM2 data using RACMO2 in the Greenland ice sheet (GrIS) and try to prove the effectiveness of CESM2 through validating GrIS SMB simulated by RACMO2 (equipped with the statistical downscaling postprocessing). This reviewer thinks that this considerable challenge is deserved to be published in the journal The Cryosphere as a brief communication if it is addressed in an appropriate manner. Overall, this paper is well written and structured; however, this reviewer suggests the following points to be considered before the publication. Please note that page and line numbers are denoted by "P" and "L", respectively.

**Specific comments (major)**

1) P. 1, L. 9-10 (and Sect. 2.1): According to the paper by Van Kampenhout et al. (2019b), which I read before reviewing this manuscript, the CESM2 simulation by Van Kampenhout et al. (2019b) was conducted following the so-called AMIP (Atmospheric Model Intercomparison Project)-run procedure. Did the authors use the same procedure/data as those presented by Van Kampenhout et al. (2019b)? If YES, the authors cannot argue "This means that, for the first time, an Earth System Model (CESM2), without assimilating observations, can be used to reconstruct historical GrIS SMB and the mass loss acceleration that started in the 1990s." in my opinion. It is because Van Kampenhout et al. (2019b) prescribe ocean and sea ice data at monthly intervals in their CESM2 simulation following the AMIP protocol, which is a kind of observation data assimilation (I mean observed ocean physical conditions can drive changes in atmospheric conditions in the model, although the atmosphere-ocean interaction would not be so strong in the model). If NO, there is no doubt that this study is amazing, and I would like to congratulate for the achievement. Anyway, please clarify this point in Sect. 2.1. No, in this work we do not use an AMIP configuration as in Van Kampenhout et al. (2019). Instead, we enable full atmosphere-ocean coupling in CESM2. This means that sea ice and sea surface temperature evolve freely. Only land ice is kept fixed. This is now clarified in the revised manuscript in P2 L27-28: "Here, we use a full atmosphere-ocean coupling in CESM2, i.e. including sea ice dynamics and sea surface temperature evolution while excluding land ice dynamics (e.g. calving)."

2) P. 6, L. 32; P. 7, L. 2: Why does CESM2-forced RACMO2.3p2 show the significant positive trend of total precipitation since 1990, which is not shown in the ERA-forced run? Please discuss. The apparent 1991-2014 positive trend in precipitation is caused by internal decadal variability in the CESM2 climate forcing. In fact, the trend becomes insignificant in the longer term (see additional figure hereunder). This is now clarified in the revised manuscript in P7 L19-20: "In addition, the positive precipitation trend disappears when extending time series using a CESM2-based SSP8.5 scenario (not shown), demonstrating that the latter trend originates from internal decadal variability."

**Specific comments (minor)**

P. 2, L. 24: If possible, please indicate/mention the GrIS ice discharge simulated by the ice sheet model CISM2.1 incorporated in CESM2, which might be of interest to readers of The Cryosphere. While CESM2 does include CISM2.1, land ice is held constant in the historical simulation. This is now clarified in P2 L27-28: "Here, we use a full atmosphere-ocean coupling in CESM2, i.e. including sea ice dynamics and sea surface temperature evolution while excluding land ice dynamics (e.g. calving)."

P. 2, L. 25-26: Please indicate data sources for "atmospheric greenhouse gas emissions (CO2 and CH4), aerosol concentrations, and land cover use". Our CESM2 simulation uses time series of atmospheric greenhouse gas and aerosol emissions and land use field following the CMIP6 standards discussed in Eyring et al. (2016). We included the reference in the revised manuscript.

P. 3, L. 4: What kind of impurities do the authors consider in RACMO2 applied in the GrIS? And, how do the authors give concentrations of the impurities in the model? Impurities consist of soot concentration that is set to 0.1 ppmv as a constant in time and space on the RACMO2 grid at 11 km. This is clarified in P3 L8-10: "In line with in situ measurements (Doherty et al., 2010), impurity concentration (soot) in RACMO2 is prescribed as a constant in time and space at 0.1 ppmv (Noël et al., 2018)."

P. 3, L. 15: Please indicate data source of sea surface temperature and sea ice extent used here (maybe from the parent CESM2 simulation results?). Indeed we prescribe the sea ice extent/cover and sea surface temperature from the CESM2 simulation. See also our response to specific comment #1. This is now clarified in P3 L19: "Sea surface temperature and sea ice extent/cover are also prescribed from the CESM2 forcing every 6 hours."

P. 3, L. 19-21: This sentence is a bit difficult to understand to me. Does it mean the 5 % lowest bare ice albedo from MCD43A3 is 0.30? This means that the 5% lowest bare ice albedo in MODIS (i.e. that can locally be lower than 0.30 for bare ice or exceed 0.55 for firn) are clipped between 0.30 and 0.55 in RACMO2. This is now reformulated in P3 L25-26 as: "[...] 5% lowest surface albedo records for the period 2000-2015, clipped between 0.30 for bare ice and 0.55 for bright [...]".

P. 5, L. 30; P. 6, L. 1: I think the results from RACMO2.1 forced by HadGEM2 is not necessary in this paper, because it can confuse readers who do not know much about RACMO2. If the authors think this part is really important for this paper, they should at least indicate key differences between RACMO2.1 and RACMO2.3p2 briefly in Sect. 2.2. Also, brief introduction of HadGEM2 would be needed as well. P. 6, L. 15-16: Same as the above comment. We deem that the comparison is valuable and shows how ESMs climate forcing has improved in time. To keep the manuscript concise and because the differences between the various RACMO2 model versions and associated forcing have been previously discussed in Van Angelen et al. (2013a,b) and Noël et al. (2015; 2018), we prefer to directly refer the reader to those publications. We included the following sentence in P6 L11-13: "For additional information about the HadGEM2-forced RACMO2.1, RACMO2.3p1 and p2 are discussed in Noël et al. (2015; 2018)."

**P. 6, L. 24: Please indicate quantitatively how realistic the simulated T700 is.**

Good suggestion. We now include T700 time series from ERA-40 (1958-1978) and ERA-Interim (1979-2014) averaged over the region  $60-80^{\circ}N$  20- $80^{\circ}W$  (black line in revised Fig. 4a hereunder). Over the period 1958-2014, T700 from "our" CESM2 member (red line) is  $0.6^{\circ}C$  colder than the reanalysis. For the ERA-Interim period only (1979-2014), the cold bias drops to  $0.4^{\circ}C$ . The recent (1991-2014) warming trend (dashed black line) is well reproduced by the CESM2 forcing (dashed red line). This is now clarified in the revised manuscript in P7 L6-10: "Compared to T700 derived from ERA-40 (1958-1978) and ERA-Interim (1979-2014; black line in Fig. 4a), the current CESM2 simulation shows a cold bias of  $0.6^{\circ}C$  over 1958-2014. For the ERA-Interim period, the bias decreases to  $0.4^{\circ}C$ . All CESM2 members show a similar warming trend after 1991, in line with the reanalysis data (dashed black line), highlighting the ability of CESM2 to represent the recent climate of Greenland. As in Fettweis et al. [...]". The figure caption has been modified accordingly.

P. 6, L. 28-29: This reviewer agrees with the authors' point that the attempt mentioned here is very interesting as long as the CESM2 data used in this study is not from the AMIP-type simulation. Please also see my first major comment. See our response to specific comment #1.

Figure 3: Can the authors briefly comment on why CESM2-forced RACMO2.3p2 could not simulate the 2012 extreme melt, which is simulated successfully by the ERA-forced run? I think this point is related to "physical drivers of the warming" (P. 6, L. 28), and any comments/suggestions by the authors will be informative for readers. Note that CESM2 does not assimilate observational data, in contrast to the reanalysis. The only forcing prescribed in CESM2 is greenhouse gas, aerosol emissions and land use cover. As a result, only the climate can be compared (e.g. the recent warming), not the weather (e.g. the 2012 melt event). In other words, CESM2-forced RACMO2 produces the right variability as e.g. expressed by extreme melt years (e.g. 2005 and 2011) that are realistic in magnitude but not necessarily in timing. This is clarified in P6 L32-33 and P7 L1: "It is important to note that, compared to forcing by reanalyses that assimilate observations, the CESM2-forced simulation produces extreme melt years (e.g. 2005 and 2011; Fig. 3b) that are realistic in magnitude but not necessarily in timing. This is clarified in magnitude but not necessarily in timing. This is clarified in magnitude but not necessarily in timing. This is clarified in magnitude but not necessarily in timing. This is clarified in P6 L32-33 and P7 L1: "It is important to note that, compared to forcing by reanalyses that assimilate observations, the CESM2-forced simulation produces extreme melt years (e.g. 2005 and 2011; Fig. 3b) that are realistic in magnitude but not necessarily in timing. This is clarified in P6 L32-33 and P7 L1: "It is important to note that, compared to forcing by reanalyses that assimilate observations, the CESM2-forced simulation produces extreme melt years (e.g. 2005 and 2011; Fig. 3b) that are realistic in magnitude but not necessarily in timing. This is clarified in P6 L32-33 and P7 L1: "It is important to note that, compared to forcing by reanalyses that assimilate observations, the CESM2-forced simulation produces extreme me

**Reviewer #2**

The authors present the results of one dynamically downscaled Earth System Model (ESM) simulation over the Greenland Ice Sheet (GrIS) and present the resulting historical surface mass balance (SMB) output from their regional climate model RACMO. After dynamical downscaling of the ESM input, the SMB is furthermore statistically downscaled to a nominal horizontal resolution of 1km. In general, the authors are doing a very good job in keeping their sentence and paragraph structure easy to follow and all their figures are well presented. Therefore, the manuscript is good to read.

**Scientific assessment** Overall, it's hard to make a case for how the study in its present form will benefit the wider cryospheric and climate community. The point of the authors here is to create a scientific foundation for additional papers that they want to write on the future contribution of the GrIS to sea level rise via (surface) mass loss. Overall, 21st century simulations of the GrIS climate and SMB would be very beneficial for the community, however, the presented analysis currently lacks the needed depth to be considered a valuable contribution to the field. Therefore, I would encourage the authors to consider the following points.

1) The authors present only one RCM simulation forced with one GCM/ESM run to create a foundation for a future paper on 21st century GrIS climate projections. However, in its current form, the paper lacks a consideration of the inter-model spread between all of the different GCMs in the CMIP5/6 model domain a consideration of how the authors made their specific selection for the one run they choose out of their CESM2 ensemble. Fettweis et al (2013) for example analyse all the CMIP5 models over the current climate, selectively find the most suitable boundary forcings and create a downscaled RCM ensemble for multiple emission scenarios and models. This point is unfortunately omitted in this study. This study assesses the ability of CESM2 (CMIP6 version) to represent the climate and SMB of the GrIS after applying dynamical (RACMO2) and statistical downscaling. The reason for choosing CESM2 is that our institute is actively involved in the improvement of the model for studies over Greenland and Antarctica, in collaboration with the National Centre for Atmospheric Research (NCAR, Boulder, USA). We have now made this motivation specific in the introduction in P2 L8-10: "The reason for selecting CESM2 as the climate forcing for RACMO2 stems from the active involvement of the Institute for Marine and Atmospheric research Utrecht (IMAU) in the development and improvement of the model for studies over both the Greenland and Antarctic ice sheets."

Of course, running multiple members of the CESM2 historical ensemble would be of added value, but doing so in a transient fashion at this high resolution is computationally prohibitive. We have now made this clear in the text in P3 L26-29: "The current study uses the climate forcing of one out of the twelve members of the CESM2 historical ensemble. Forcing RACMO2 with other CESM2 members would have been ideal, but doing so in a transient fashion and at high spatial and temporal resolution is computationally prohibitive. Instead, we select one member that offers the 6-hourly climate forcing required to drive RACMO2 while being representative of other CESM2 members (see Section 4.3 and Fig. 4a)."

This ensemble member was selected because it had 6-hourly forcing available and is representative of other members; Fig. 4 shows that there is no reason to believe the results would be different if another member had been chosen. Based on these considerations, we judge that our conclusions on the quality of the CESM2 climate forcing are robust.

2) The authors focus their analysis only on the GrIS surface mass balance If this study should become a standalone piece of work without the promised future projections, then the authors should be highly encouraged to consider at least a subset of other parameters to validate their single-simulation analysis to exclude the likelihood of compensating biases leading to a "correct" SMB due to "false" physical reasons - (a) Surface energy budget vs. observations (b) Albedo vs. observations (c) Temperature and/or cloud properties vs. observations. We decided to limit the evaluation to SMB measurements, as the ability of CESM2 to represent key surface processes (including the near surface climate and the surface energy budget, SEB) has been addressed in other recent publications that emerged from the CESM2 development phase, e.g. Van Kampenhout et al. (2019) and Sellevold et al. (2019). In addition, direct comparison to daily in situ measurements (e.g. PROMICE, GC-NET) of (a) SEB components, (b) snow albedo, (c) near-surface temperature and cloud properties is not appropriate since ESMs, as opposed to reanalysis, do not assimilate observations and hence cannot reproduce the actual weather and exact timing of extremes (as in e.g. 2010 and 2012). See also our response to Reviewer #1 on Figure 3. We therefore deem the good agreement with in situ SMB measurements in different regions of the GrIS, characterized by very different climate conditions, to be a solid model evaluation, especially in view of the excellent agreement with temporal mass loss from GRACE.

3) If the reader assesses the novelty based on what the authors highlight "...for the first time an ESM (CESM2) can be used to reconstruct historical SMB..." then the science of the paper would need to be judged either on the claim (a) that is "the first time" or (b) that the "historical SMB" is more accurate than from other model setups. We have chosen option (a), as to the authors' knowledge no ESM-forced RCM simulation has ever accurately simulated the SMB before the 1990s and reproduced the post-1991 mass loss in close agreement with GRACE. We point out that Reviewer #1 agrees with this: "If NO, there is no doubt that this study is amazing, and I would like to congratulate for the achievement."

4) However, (a) e.g. Fettweis et al. (2013) as a benchmark already show that GCMs/ESMs can be used to force RCMs over the historical period and roughly get the magnitude of the SMB components right. (b) The most accurate "historical SMB" does not come from this model setup, but rather from regional climate models that downscale observation-based reanalysis data (e.g. RCM with ERA-I or ERA-5). The presented results (Figure 3) unsurprisingly show that CESM2-RACMO does not capture the interannual SMB variability and extremes (e.g. melt in 2012) which is expected with GCM boundary forcings. However, it means that the accuracy of historical SMB representation is also not an advancement of the scientific knowledge. The fact that no additional bias

correction in the forcing field is required to obtain accurate SMB is novel. We also disagree with the statement that "the accuracy of historical SMB representation is also not an advancement of the scientific knowledge". The reduced uncertainty in historical SMB reconstruction from ESM forcing as shown here is the **only way** to assess the reliability of future climate projections.

**Recommendations** The reviewer would like to encourage the authors to either add significant extra analysis to their current model and study setup to create a solid foundation for their promised future attribution studies, or potentially add the presented analysis to their upcoming future projections altogether. The authors could potentially consider some of the following points/questions when considering the next steps for their analysis post-review.

1) Given the limited amount of future GrIS mass loss studies with RCMs and GCM forcing, the scientific interest of the presented approach lies in the actual future projections, not necessarily on the historical SMB reconstructions due to obvious limitations when using GCM/ESM boundary conditions. Please see our previous responses to scientific assessment #1, 3 and 4.

2) How representative is this one CESM2 run compared to the spread in CMIP5/6 simulations? Other recent studies have found great uncertainties in future GrIS projections using RCMs to downscale GCMs/ESMs which is/are not really discussed yet in the manuscript. What if the authors would force RACMO with other GCMs? How well does the current setup represent the surface energy budget, temperature, albedo, cloud properties? Please see our previous responses to scientific assessment #1 and 2. In addition, assessing uncertainties in future projections is beyond the scope of this study that focuses on the ability of the CESM2 climate forcing to represent the present-day SMB of the GrIS.

3) If forcing RACMO with other GCMs is technically not feasible, then one approach would be to force RACMO with additional ensemble members presented in Figure 4. The robustness of the SMB and potential underlying compensating errors can hardly be assessed by only one simulation. Please see our previous response to scientific assessment #1.

**Minor comments**

P1.L9: "without assimilating observations" is this correct? The methods of the paper claim that RACMO uses satellite albedo to constrain the surface albedo. Please clarify. Good point, of course we meant that **CESM2** is not constrained by observations. This is now clarified in P1 L9-11 as follows: "This means that, for the first time, climate forcing from an Earth System Model (CESM2), that assimilates no observations, can be used without additional corrections to reconstruct historical GrIS SMB [...]".

P3.L19: "bare ice albedo is prescribed from ... MODIS.." – please see first minor comment and clarify. P3.L28 Also in the statistical downscaling technique the authors use observed MODIS albedo. Please see the first comment on how this fits with the claim that this study doesn't use assimilated observations. Please see the answer above in P1 L9.

P3.L32-33: Does it only change the runoff and SMB or also improve the statistical comparison? Statistical downscaling aims at resolving narrow marginal glaciers, ablation zones, and associated large SMB gradients not resolved by the 11 km grid, as well as correcting for the bare ice albedo bias in RACMO2. As a result, statistical downscaling primarily increases marginal runoff, which improves the SMB agreement with observations. The method is presented in detail in Noël et al. (2016).

P4.L24: "due to the high quality of the CESM2 climate" but also e.g. P1.L5 "good comparison" and P5.L6 "shows excellent agreement" and at other points in the manuscript - these are quite colloquial expressions with little scientific meaning. What does a "high quality" climate in a GCM mean? The manuscript doesn't even currently evaluate the CESM2 climate for example. Good point. In P1 L5 we deem that evaluation statistics should not be listed in the abstract, the "good agreement" and associated statistics are elaborated in more detail in Sections 3 and 4. In P5 L6 we feel that mass loss derived from combined observed ice discharge and modelled SMB of 3,299 Gt yr-1 is indeed in "excellent agreement" with GRACE estimates of 3,290 Gt yr-1. Concerning the "high quality" statement, we decided to remove the sentence in P4 L23-25.

P4.L25ff: But what about other parameters such as the surface energy budget, temperature and clouds? How does it compare to recent circulation and cloud anomalies over Greenland which have been shown to be important for future projections? Upper atmospheric temperature (T700) in the CESM2 forcing is now evaluated using ECMWF reanalyses in Fig. 4a. See also our response to scientific assessment #2. Addressing circulation and cloud anomalies is beyond the scope of this study: this work assesses the ability of the CESM2 climate forcing to reconstruct the present-day SMB of the GrIS.

P5.L6-8: The acceleration (i.e. dSMB/dt) is likely not discussed here but rather a "total mass loss". Thank you for pointing this out, we meant "mass loss" rather than "mass loss acceleration". This is now corrected in P5 L14 as follows: "[...] realistically capture the recent Greenland mass loss (2003-2014) (Bamber et al., 2018)."

P5.L30-32ff: ad HadGEM; "did not accurately reproduce SMB". a) Throughout this study the reader is often left in the dark as to "Why?" certain numbers or results are mentioned, and why certain processes behave the way they do. At the moment, the paper is an ensemble of nice figures and easy-to-follow text, but the study and the reader would highly benefit if the authors would more often dig into the question of "Why?" some processes and numbers are reported here and apparently deemed important for the reader. b) This would also be a good point to address the matter why HadGEM and CESM2 produce such different SMB/ME/RU results (+-50%)? Is it due to differences in the lateral forcings/ the internal RACMO physics/ circulation / cloud physics? Hofer et al. (2019) for example show the large spread in GrIS SMB that can result from different GCM forcing. To address the latter, we now include this sentence in P6 L26-28: "The reason is that, unlike CESM2 (Van Kampenhout et al., 2019a), the HadGEM2 forcing had a strong, systematic warm bias of ~1ºC (Van Angelen et al., 2013a), resulting in overestimated meltwater runoff and thus underestimated SMB (Fig. 2d)." Regarding the first comment, the topic of this paper is how development of CESM2 in particular (and therewith ESMs in general) has led to much improved representation of (downscaled) GrIS SMB. Back in its time, HadGEM2 was too warm over Greenland and required corrections to obtain an acceptable GrIS SMB. The main message of this paper is that this kind of corrections is now no longer required and even recent (mass loss) trends are captured correctly. We feel that for a short communication, this presents sufficient advance of the state-ofthe-art to warrant publication. At the same time, we deem that it is beyond the scope of this paper to analyse problems in a -now obsolete- GCM.

P7.L8-9 "can reliably reproduce ... variability of historical SMB" – When looking at Figure 3 the GCM forced SMB reconstruction clearly lacks the ability to reproduce the interannual SMB variability and extremes shown in Figure 3A when RACMO is forced by reanalysis. Just as an example, the extreme melt summer of 2012 accurately captured in Figure 3A is not present in Figure 3B, therefore the reader considers this to be a doubtful assumption. See our response to Reviewer #1 on Figure 3.

P7.L3-4: unclear phrasing "is for 60%" Thank you, this has been reformulated as follows in P7 L20: "[...] Fig. 3c shows that 60% of the recent mass loss acceleration in CESM2-forced RACMO2.3p2 is caused by decreased SMB [...]".

P7.L7-10: What are the uncertainties coming from the lack of a multi model forcing (e.g. Fettweis et al. (2013). See our response to scientific assessment #1.

Figure 1: How does it compare during melt season? How does the SEB compare to the observational networks of PROMICE, DMI and/or GCNET? See our response to scientific assessment #2.

Figure 2: Please clarify the choice of HadGEM and not other GCMs? If it is feasible to force RACMO with other GCMs then please consider analysing the intermodel spread of the GrIS climate when RACMO is forced by other GCMs. See our response to scientific assessment #1.

**Brief communication: CESM2 climate forcing (1950-2014) yields realistic Greenland ice sheet surface mass balance**

Brice Noël1, Leonardus van Kampenhout1, Willem Jan van de Berg1, Jan T. M. Lenaerts2, Bert Wouters1,3, and Michiel R. van den Broeke1

[revised manuscript text omitted]

---

## Referee Report (RR1)

**I only focus on changes asked by the reviewers and that authors chose not to take into account. My comments are in bold.**

Reviewer#1

P. 5, L. 30; P. 6, L. 1: I think the results from RACMO2.1 forced by HadGEM2 is not necessary in this paper, because it can confuse readers who do not know much about RACMO2. If the authors think this part is really important for this paper, they should at least indicate key differences between RACMO2.1 and RACMO2.3p2 briefly in Sect. 2.2. Also, brief introduction of HadGEM2 would be needed as well. P. 6, L. 15-16: Same as the above comment. We deem that the comparison is valuable and shows how ESMs climate forcing has improved in time. To keep the manuscript concise and because the differences between the various RACMO2 model versions and associated forcing have been previously discussed in Van Angelen et al. (2013a,b) and Noël et al. (2015; 2018), we prefer to directly refer the reader to those publications. We included the following sentence in P6 L11-13: "For additional information about the HadGEM2-forced RACMO2.1 simulation and settings, we refer the reader to Van Angelen et al. (2013a); key differences between RACMO2.1, RACMO2.3p1 and p2 are discussed in Noël et al. (2015; 2018)."

**I agree that RACMO2.1 forced by HadGEM2 is not necessary in this paper. We cannot know if the difference is coming from RACMO2.1, the downscaling method to 1 km, of HadGEM2 vs. CESM2, whereas only the later is of interest for this article. I think that Fig. 2 is not useful and should be removed, on focus only on the difference between 11 km and 1 km, as SMB components are shown in Fig. 3.**

Reviewer#2

2) The authors focus their analysis only on the GrIS surface mass balance If this study should become a standalone piece of work without the promised future projections, then the authors should be highly encouraged to consider at least a subset of other parameters to validate their single-simulation analysis to exclude the likelihood of compensating biases leading to a "correct" SMB due to "false" physical reasons - (a) Surface energy budget vs. observations (b) Albedo vs. observations (c) Temperature and/or cloud properties vs. observations. We decided to limit the evaluation to SMB measurements, as the ability of CESM2 to represent key surface processes (including the near surface climate and the surface energy budget, SEB) has been addressed in other recent publications that emerged from the CESM2 development phase, e.g. Van Kampenhout et al. (2019) and Sellevold et al. (2019). In addition, direct comparison to daily in situ measurements (e.g. PROMICE, GC-NET) of (a) S EB components, (b) snow albedo, (c) near-surface temperature and cloud properties is not appropriate since ESMs, as opposed to reanalysis, do not assimilate observations and hence cannot reproduce the actual weather and exact timing of extremes (as in e.g. 2010 and 2012). See also our response to Reviewer #1 on Figure 3. We therefore deem the good agreement with in situ SMB measurements in different regions of the GrIS, characterized by very different climate conditions, to be a solid model evaluation, especially in view of the excellent agreement with temporal mass loss from GRACE.

**I agree with reviewer#2 that given that the aim of the article is to validate CESM2, the authors have to show that CESM2-RACMO2 SMB is right for the right reason. The authors could focus on evaluating the forcing fields of RACMO2 compared to ERA-40 and ERA-Interim in a climatological point of view, for all ensemble members, similarly as done in Fig. 4, adding more variables such as T850, Z500. In particular, is the increase in temperature in**

**CESM2 ensemble related to circulation change after the 90's as observed (Hanna et al. 2018, cited by the authors in the introduction), or to high climate sensitivity?**

**Looking at separate surface energy balance components as suggested by reviewer#2, in a climatological point of view, can also help answer if RACMO2-CESM2 and RACMO2-ERA are increasing melt after the 90s for the same reason.**

P4.L25ff: But what about other parameters such as the surface energy budget, temperature and clouds? How does it compare to recent circulation and cloud anomalies over Greenland which have been shown to be important for future projections? Upper atmospheric temperature (T700) in the CESM2 forcing is now evaluated using ECMWF reanalyses in Fig. 4a. See also our response to scientific assessment #2. Addressing circulation and cloud anomalies is beyond the scope of this study: this work assesses the ability of the CESM2 climate forcing to reconstruct the present-day SMB of the GrIS.

**I disagree with the authors and agree with reviewer#2. I can't see why assessing clouds and circulation changes is beyond the scope of this study. It is the core of this study to assess whereas CESM2 is doing the right SMB for the right reason.**

---

## Author Response (AR2)

**Response to editor**

Dear editor we would like to thank you for giving us the opportunity to revise our manuscript. To facilitate readability, our responses to your remaining comments are displayed in blue and modifications in the manuscript are highlighted in red. Minor textual corrections are also displayed in red in the revised manuscript.

**Editor comments:**
As you know well, in most of my papers using MAR forced by GCMs (TC, 2013) or reanalyses (TC, 2017), I discussed the reliability of the MAR forcings because the trend coming RCM is fully driven by the forcing and therefore it is important for me to evaluate a bit the forcing to interpret the RCM results. Therefore, I disagree in part with you when you claim that discussing a bit the CESM2 ability to simulate the recent general circulation changes vs the high CESM2 climate sensitivity is beyond the scope of this paper. However, as a large part of you are involving in the CEMS2 development, I agree that this can be discussed in another paper focusing on CESM2 alone and I am therefore OK to accept your paper (with a final review only from me) if you:

1) Focus more on the ability of CESM2-RACMO to simulate the mean contemporary SMB of the Greenland ice sheet without, as you have highlighted, any assimilation of observations or biases correction (as I did when MAR has been forced by the 20th century reanalyses). The manuscript has been revised accordingly to focus on the historical GrIS SMB reconstruction. We hope that our textual modifications better reflect this and clarify our main objective.

2) Adding HadGEM2-RACMO2.1 in the discussion is then justified as basis of comparison. We now provide a justification for why the HadGEM2-forced RACMO2.1 simulation was used for comparison, and included a description of the model, simulation settings and main differences with the current CESM2-forced RACMO2.3p2 simulation. In L28 of P5 to L6 of P6, we inserted: "To highlight recent improvements in ESM climate forcing, we use the modelled SMB components from a previous RACMO2.1 simulation forced by HadGEM2 (dark green dots). HadGEM2 is a coupled atmosphere-ocean model developed by the Met Office Hadley Center using a spatial resolution of 1.875º x 1.25º (atmosphere) and 1º x 1º (ocean). Similar to CESM2, the model prescribes land cover use, greenhouse gas and aerosol emissions, and simulates the dynamic evolution of sea ice and sea surface temperature over the historical period (Jones et al., 2011). Compared to RACMO2.1, the latest RACMO2.3p2 version implements 1) significant changes in the cloud physics favouring more snowfall in the ice sheet interior; 2) lower impurity concentration in snow (soot) and smaller snow grain size, both reducing the previously underestimated snow albedo; 3) less active snow drift erosion limiting the overestimated exposure of bare ice notably in the northeast of Greenland. For additional information about the HadGEM2-forced RACMO2.1 simulation and settings, we refer the reader to Van Angelen et al. (2013a); key differences between RACMO2.1, RACMO2.3p1 and p2 are discussed in Noël et al. (2015, 2018)."

3) If you still discuss the comparison with the GRACE trend, that you add at least a small sentence like the one below: "It is important to note that the main drivers of the recent mass loss over the 2000s from RACMO-ERA and RACMO-CESM2 are likely different since no GCM from CMIP5 or CMIP6 (eg: Hanna et al., TC, 2018; Delhasse et al., TCD, 2020) project the recent observed summer circulation changes, known to have driven in part the recent mass loss (Noel et al., Science Advances, 2019)." This leaves the opportunity to the CESM2 guys to discuss this issue more in details but in the same time, this allows to be honest with the TC readers (e.g. CESM2 is one of the model used in Delhasse et al., 2020) and the ones who will use your model outputs.
To tone down the comparison with GRACE-derived mass loss trend, we reformulated:

[revised manuscript text omitted]

---

## Author Response (AR3)

**Response to editor**

Dear editor, you can find hereunder our response to your remaining comment. Modifications in the manuscript are highlighted in red.

**Editor comment:**
As announced by email, I have reviewed it myself and I'm happy to accept it for final publication in TC. While it is already mentioned pg 8 lines 2-5, I only request that you remember to readers, somewhere in your conclusion (Section 5), the issue about the recent general circulation changes not simulated by ESMs to be fair with the reviewers' concerns about this.

You could for example add something like this ", although no ESMs from CMIP5 (and likely CMIP6) accurately reproduces the recent observed changes in summertime Arctic atmospheric circulation, driving in part the recent mass loss" at the end of the sentence, line 15, pg 8.
We modified the conclusions accordingly and included the following sentence in L22-24 of P8: "
[revised manuscript text omitted]